# Mutation Spectrum of *GJB2* in Taiwanese Patients with Sensorineural Hearing Loss: Prevalence, Pathogenicity, and Clinical Implications

**DOI:** 10.3390/ijms26178213

**Published:** 2025-08-24

**Authors:** Yi-Feng Lin, Che-Hong Chen, Chang-Yin Lee, Hung-Ching Lin, Yi-Chao Hsu

**Affiliations:** 1Institute of Biomedical Sciences, Mackay Medical University, New Taipei City 252005, Taiwan; linfong5202@gmail.com; 2Department of Audiology and Speech-Language Pathology, Mackay Medical University, New Taipei City 252005, Taiwan; hclin59@mmu.edu.tw; 3Department of Chemical and Systems Biology, Stanford University School of Medicine, Stanford, CA 94305-5101, USA; chehong@stanford.edu; 4Department of Chinese Medicine, E-DA Hospital, Kaohsiung 82445, Taiwan; michael11556688@gmail.com; 5Department of Chinese Medicine, E-DA Cancer Hospital, Kaohsiung 82445, Taiwan; 6College of Medicine, The School of Chinese Medicine for Post Baccalaureate, I-Shou University (Yancho Campus), Kaohsiung 82445, Taiwan; 7Department of Otolaryngology, Mackay Memorial Hospital, Taipei 104217, Taiwan

**Keywords:** non-syndromic hearing loss, *GJB2*, gene mutations

## Abstract

Hearing loss is often caused by genetic and environmental factors, with inherited mutations responsible for 50–60% of cases. The *GJB2* gene, encoding connexin 26, is a major contributor to nonsyndromic sensorineural hearing loss (NSHL) due to its role in cellular communication critical for auditory function. In Taiwan, common deafness-associated genes include *GJB2*, *SLC26A4*, *OTOF*, *MYO15A*, and *MTRNR1*, which were similar to those found in other populations. The most common pathogenic genes is *GJB2* mutations and the hearing level in children with *GJB2* p.V37I/p.V37I or p.V37I/c.235delC was estimated to deteriorate at approximately 1 decibel hearing level (dB HL)/year. We found another common mutation in Taiwan Biobank, *GJB2* p.I203T, which were identified in our data and individuals carrying this mutation experienced more severe hearing loss, suggesting a synergistic effect of these mutations on auditory impairment. We suggest *GJB2* whole genetic screening is recommended for clinical management and prevention strategies in Taiwan. This study used data from the Taiwan Biobank to analyze allele frequencies of *GJB2* gene variants. Predictive software (PolyPhen-2 version 2.2, SIFT for missense variants 6.2.1, MutationTaster Ensembl 112 and Alphamissense CC BY-NC-SA 4.0) assessed the pathogenicity of specific mutations. Additionally, 82 unrelated NSHL patients were screened for mutations in these genes using PCR and DNA sequencing. The study explored the correlation between genetic mutations and the severity of hearing loss in patients. Several common *GJB2* mutation sites were identified from the Taiwan Biobank, including *GJB2* p.V37I (7.7%), *GJB2* p.I203T (6%), *GJB2* p.V27I (31%), and *GJB2* p.E114G (22%). Bioinformatics analysis classified *GJB2* p.I203T as pathogenic, while *GJB2* p.V27I and *GJB2* p.E114G were considered polymorphisms. Patients with *GJB2* p.I203T mutation experienced more severe hearing loss, emphasizing the potential interaction between the gene in auditory impairment. The mutation patterns of *GJB2* in the Taiwanese population are similar to other East Asian regions. Although GJB2 mutations represent the predominant genetic cause of hereditary hearing loss, the corresponding mutant proteins exhibit detectable aggregation, particularly at cell–cell junctions, suggesting at least partial trafficking to the plasma membrane. Genetic screening for these mutations—especially *GJB2* p.I203T (6%), *GJB2* p.V27I (31%), and *GJB2* p.E114G (22%)—is essential for the effective diagnosis and management of non-syndromic hearing loss (NSHL) in Taiwan. We found *GJB2* p.I203T which were identified in our data and individuals carrying this mutation experienced more severe hearing loss, suggesting a synergistic effect of these mutations on auditory impairment. We suggest whole *GJB2* gene sequencing in genetic screening is recommended for clinical management and prevention strategies in Taiwan. These findings have significant clinical and public health implications for the development of preventive and therapeutic strategies.

## 1. Introduction

Hearing loss affects 1.5 billion people worldwide, including 34 million children, with 60% of cases being preventable. Congenital hearing loss is a major focus in medical research due to its prevalence, affecting 1–2 out of every 1000 newborns (https://www.who.int/news-room/fact-sheets/detail/deafness-and-hearing-loss, accessed on 26 February 2025). Approximately 70% of these cases are classified as non-syndromic hearing loss (NSHL). Mutations in the *GJB2*, *SLC26A4*, and *MTRNR1* genes (the latter encoding mitochondrial 12S rRNA) are frequently associated with NSHL, *GJB2* mutations alone responsible for up to 50% of autosomal recessive NSHL cases. The condition exhibits significant genetic heterogeneity [1].

The *GJB2* gene is frequently mutated in individuals with hearing impairment in the world (Figure 1). Recessive *GJB2* variants, the most common genetic cause of hearing loss, may contribute to progressive NSHL [2]. To determine the rate of progression of SNHI with the *GJB2* p.V37I/p.V37I and *GJB2* p.V37I/c.235delC genotypes, they performed linear regression analysis of the relationship between the follow- up time (years) and the hearing level (dB HL). In the past study, the most common mutation in Taiwan is *GJB2* p.V37I homozygous mutation, the prevalence of *GJB2* is about 21.7%. In the Taiwanese population, the most common deafness mutations are *GJB2* c.109G>A (p.V37I) and *GJB2* c.235delC (p.L79CfsTer3) [3,4]. The rate of SNHI progression in children with *GJB2* p.V37I/p.V37I did not significantly differ from that in children with *GJB2* p.V37I/c.235delC. When pooled together, the hearing level in children with *GJB2* p.V37I/p.V37I or *GJB2* p.V37I/c.235delC was estimated to deteriorate at approximately 1 dBHL/year (hearing level (dB HL) = 18.7 + 1.0 × year). They delineated the longitudinal auditory features of the highly prevalent *GJB2* p.V37I mutation on a general population basis and confirmed the utility of newborn genetic screening in identifying infants with late-onset or progressive hearing impairment undetectable by newborn hearing screening [5]. Beyond these mutations, the Taiwan Biobank has identified other prevalent single nucleotide polymorphisms (SNPs) among Taiwanese people, including *GJB2* c.608T>C (p.I203T) at 6.1%, *GJB2* c.79G>A (p.V27I) at 31%, and *GJB2* c.341A>G (p.E114G) at 22% (Table 1). Interestingly, Taiwan shows a much higher incidence of the *GJB2* p.I203T SNP compared to Japan (1%), South Korea (2%), and China (3%) [6,7,8]. Despite its high frequency, research on this particular SNP is limited, underscoring the need for more in-depth studies. Regarding specific mutations, the *GJB2* c.109G>A (p.V37I) missense mutation, located in the M1 transmembrane domain, is a missense mutation likely involved in connexon assembly [9]. Evidence from transfected HeLa cells indicates that this GJB2 mutation results in plaque-like distribution reminiscent of wild-type Cx26 expression. Nonetheless, the presence of prominent aggregates, especially at cell–cell interfaces, implies aberrant yet partial trafficking of the mutant protein to the plasma membrane [10]. Conversely, the *GJB2* c.235delC (p.L79CfsTer3) mutation, a frameshift mutation in the M2 transmembrane domain, results in premature translation termination at codon 81, producing a polypeptide 145 amino acids shorter than wild-type Cx26, thereby preventing the formation of functional gap junctions.

In the Taiwanese population, the *GJB2* p.V37I (7.7%), *GJB2* p.I203T (6%), *GJB2* p.V27I (31%), and *GJB2* p.E114G (22%) mutations represent frequently genetic alterations. Despite the extensive research on the individual roles of this mutations associate with various diseases, there remains a paucity of comprehensive studies examining the combined effects of these mutations on auditory function [11]. The underlying molecular mechanisms and the potential therapeutic strategies for hearing loss associated with these genetic mutations warrant further investigation [12].

In the present study, we aim to investigate the genotype-phenotype correlation of the four missense mutations with highest allele frequency of *GJB2* gene in Taiwanese population. This study will increase the understanding of the genetic functional implications of hearing loss-related genes and clarify the pathogenesis of *GJB2* related hearing loss and provide new insights and solutions for its prevention and protection.

## 2. Results

### 2.1. Global and Taiwanese Prevalence of GJB2 Variants

To establish a broader context for interpreting our findings, we first compiled data on the prevalence of *GJB2* variants globally and compared it with data from the Taiwanese population (Figure 1). This comparative overview revealed notable differences in variant frequencies between populations, underscoring the unique genetic architecture of *GJB2*-related hearing loss in Taiwan.

### 2.2. Structural and Pathogenic Insights into Common GJB2 Missense Mutations

We next performed a structural and in silico pathogenicity analysis of several *GJB2* missense mutations that are prevalent in Taiwan (Figure 2). Among these, the *GJB2* p.I203T (c.608T>C) variant, situated in the fourth transmembrane domain (M4), demonstrated an AlphaMissense score of 0.646 and was classified as likely pathogenic. Due to its location within the membrane-spanning region, this substitution is hypothesized to compromise the conformational stability of connexin 26, potentially disrupting gap junction assembly or altering ionic permeability.

**Figure 2 ijms-26-08213-f002:**
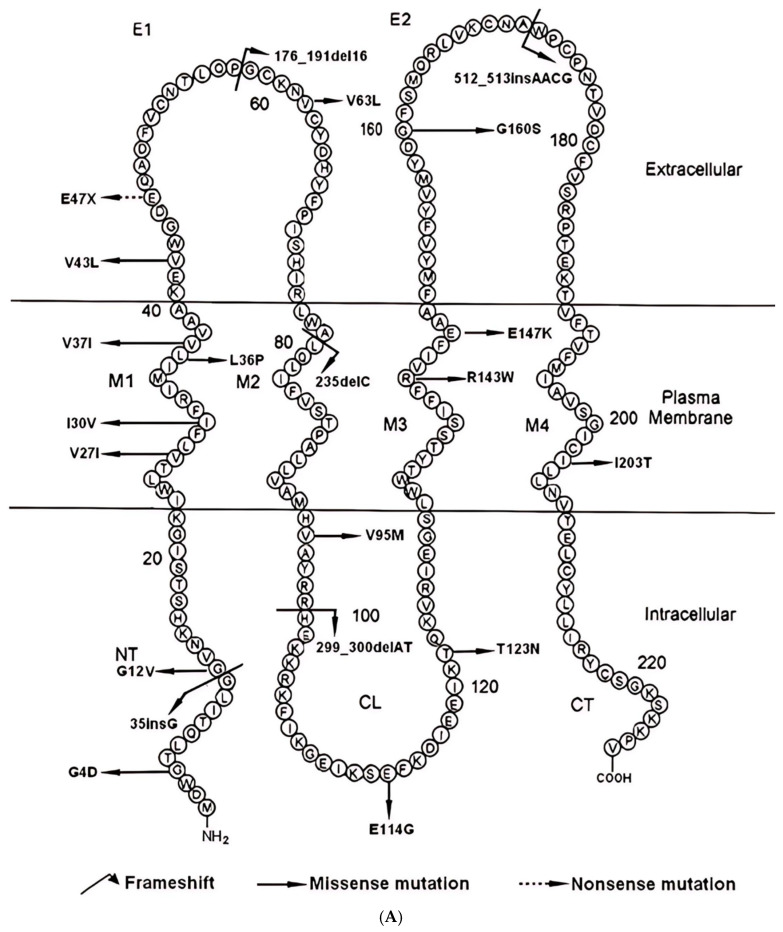
(**A**) Currently known distribution of *GJB2* gene mutations. Schematic representation of the human GJB2 (connexin 26) protein topology and reported mutations. The diagram depicts the four transmembrane domains (M1–M4), two extracellular loops (E1, E2), one intracellular loop (CL), and the N- and C-terminal regions. Amino acid substitutions and sequence alterations are mapped to their respective positions, with mutation types indicated by different arrow styles and colors: frameshift mutations, missense mutations, and nonsense mutations. Numbering corresponds to the amino acid positions in the GJB2 protein sequence [13]. (**B**) Currently known distribution of *GJB2* gene missense mutations. To comprehend the potential reasons for varying degrees of *GJB2*-related hearing loss, we plotted the positions of *GJB2* mutations and identified that p.I203T is situated within the transmembrane segment towards the C-terminus. By utilizing online software to visualize its 3D structural depiction (Figure 3), we observed that the amino acid alteration from the nonpolar, hydrophobic isoleucine to the uncharged, hydrophilic threonine characterizes the structural change induced by p.I203T and the negative, hydrophilic glutamic acid to the nonpolar, hydrophobic glycine characterizes the structural change induced by p.E114G. There is no change in the amino acid properties of p.V27I and p.V37I, both of them are from valine to isoleucine, nonpolar and hydrophobic. Most literature still classify *GJB2* p.I203T as a benign variant. We shall see if it is more or less like p.V37I due to low penetrance. Based on Table 2, our 23 patient p.I203T variant frequency is 3/46 alleles = 6.5%, similar to Taiwan Biobank 1517 healthy subjects data. However, p.V37I frequency in 23 patients was clearly higher than Taiwan Biobank (7/46 = 15.2%), this include one patient with homozygous p.V37I mutation.

**Table 2 ijms-26-08213-t002:** Missense mutations of the *GJB2* gene in the Taiwanese population, with p.V27I (c.79G>A), p.V37I (c.109G>A), p.E114G (c.341A>G), and p.I203T (c.608T>C) representing the four most prevalent pathogenic variants identified in clinical and population-based studies.

GRCh37 (hg19)	ID	Domain	cDNA Position	Amino Acid Substitution	Polyphen2	SIFT	Mutationaster	Alphamissense	Freq
chr13:20763710	rs111033222	IC1	c.11G>A	p.G4D	Ben (0.029)	TOLERATED (0.21)	polymorphism	benign	C:0.996042 T:0.003958
chr13:20763686	rs1801002	IC1	c.35G>A	p.G12D	Dam (0.999)	TOLERATED (0.45)	DC (0.999)	Pathogenic (0.8012)	C:0.999658 A:0.000342
chr13:20763642	rs2274084	TM1	c.79G>A	p.V27I	Dam (0.999)	TOLERATED (0.13)	polymorphism	benign	C:0.692815 T:0.307185
chr13:20763633	rs374625633	TM1	c.88A>G	p.I30V	Ben (0.226)	TOLERATED (0.31)	DC (0.999)	benign	T:0.998022 C:0.001978
chr13:20763614		TM1	c.107T>C	p.L36P	Dam (1.000)	AFFECT PROTEIN FUNCTION (0.00)	DC (0.999)	Pathogenic (0.991)	A:0.999670 G:0.000330
chr13:20763612	rs72474224	TM1	c.109G>A	p.V37I	Dam (1.000)	TOLERATED (0.66)	DC (0.999)	benign	C:0.923204 T:0.076796
chr13:20763534		EC1	c.187G>A	p.V63M	Dam (1.000)	AFFECT PROTEIN FUNCTION (0.00)	DC (0.999)	Pathogenic (0.817)	C:0.999670 A:0.000330
chr13:20763380	rs2274083	IC2	c.341A>G	p.E114G	Ben (0.001)	TOLERATED (0.32)	polymorphism	benign	T:0.778034 C:0.221966
chr13:20763353	rs111033188	IC2	c.368C>A	p.T123N	Ben (0.000)	TOLERATED (0.51)	polymorphism	benign	G:0.991760 T:0.008240
chr13:20763341		IC2	c.380G>A	p.R127H	Ben (0.006)	TOLERATED (0.22)	polymorphism	benign	C:0.999341 A:0.000659
chr13:20763264	rs111033186	EC2	c.457G>A	p.V153I	Ben (0.002)	TOLERATED (1.00)	DC (0.815)	benign	C:0.999670 T:0.000330
chr13:20763243	rs34988750	EC2	c.478G>A	p.G160S	Dam (0.947)	AFFECT PROTEIN FUNCTION (0.02)	DC (0.999)	benign	C:0.999670 T:0.000330
chr13:20763150		EC2	c.571T>C	p.F191L	Dam (1.000)	AFFECT PROTEIN FUNCTION (0.00)	DC (0.999)	Pathogenic (0.997)	A:0.997034 G:0.002966
chr13:20763113	rs76838169	TM4	c.608T>C	p.I203T	Dam (0.994)	AFFECT PROTEIN FUNCTION (0.01)	DC (0.999)	Pathogenic (0.646)	A:0.939024 G:0.060976

IC intracellular, TM transmembrane, EC extracellular. The amino acid substitution is predicted damaging by the score of PolyPhen-2 (0 being least and 1 being most), SIFT (ranges from 0 to 1, damaging ≤ 0.05, tolerated > 0.05) Mutationtaster and Alphamissense.

In contrast, the *GJB2* p.E114G (c.341A>G) variant, located in the intracellular loop between the second and third transmembrane domains (M2–M3), exhibited a much lower AlphaMissense score of 0.128 and was predicted to be likely benign. Although this domain may contribute to intracellular signaling or regulatory functions of the channel, the substitution of a negatively charged glutamate with a neutral glycine is unlikely to induce significant structural or functional disruptions.

The two most prevalent *GJB2* variants in the Taiwanese population, *GJB2* p.V37I (c.109G>A) and *GJB2* p.V27I (c.79G>A), both reside within the first transmembrane domain (M1) and have similarly low AlphaMissense scores of 0.156 and 0.111, respectively. These variants are commonly associated with mild-to-moderate hearing loss in homozygous or compound heterozygous states. However, given that the amino acid substitutions involve conservative changes from valine to isoleucine, they are predicted to exert minimal impact on protein folding or channel function, particularly in heterozygous individuals. Collectively, these findings emphasize the necessity of integrating allele frequency data, computational pathogenicity predictions, and structural localization to more accurately assess the clinical relevance of *GJB2* missense variants (Figure 3).

**Figure 3 ijms-26-08213-f003:**
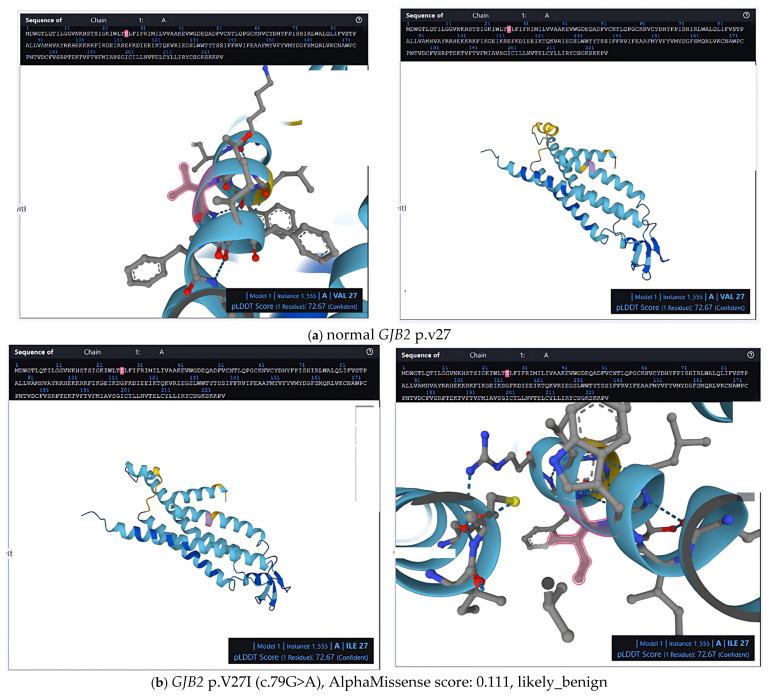
3D protein structure map of human normal *GJB2* and *GJB2* mutations. The predicted structure by ESM fold (https://esmatlas.com/resources?action=fold, accessed on 18 August 2024) is colored by local prediction confidence (pLDDT) per amino acid location. Blue indicates confident predictions (pLDDT > 0.9), while red indicates low confidence (pLDDT < 0.5).

### 2.3. Population-Level Analysis of Missense Mutations in the Taiwanese Cohort

To further investigate the mutation spectrum of *GJB2* in the general population, we analyzed data from 1517 healthy individuals in the Taiwan Biobank. A complete listing of all identified missense mutations, along with their predicted pathogenicity scores from PolyPhen-2, SIFT, and MutationTaster, is summarized in Table 2.

The analysis of *GJB2* confirmed the high prevalence of several key variants, including p.V27I (c.79G>A) at 29.7%, p.E114G (c.341A>G) at 21.2%, and p.I203T (c.608T>C) at 6%.

The *GJB2* p.V37I (c.109G>A) mutation remained the most commonly detected variant overall, though its precise frequency was not reiterated in this specific analysis. These frequencies provide crucial insights into the carrier rates and potential clinical implications of these variants in an ostensibly healthy Taiwanese population.

### 2.4. Clinical Implications and Observed Genotype–Phenotype Correlations

Several genotype–phenotype associations emerged from our analysis. The *GJB2* p.I203T variant, in particular, was frequently linked to profound hearing loss, consistent with its likely pathogenic classification and structural positioning. Meanwhile, variants such as *GJB2* p.V27I and *GJB2* p.E114G were often detected in compound heterozygous states, co-occurring with other *GJB2* mutations in 56.5% and 47.8% of cases, respectively. These variants were rarely found in isolation, suggesting potential synergistic or additive effects in contributing to the hearing phenotype.

Furthermore, we observed that the presence of additional mutations in other deafness-associated genes, such as *SLC26A4* and *OTOF*, was associated with more severe auditory impairment. These findings support a multifactorial model of hearing loss, in which both individual variant pathogenicity and complex genetic interactions play a role in determining clinical severity.

Combining the aforementioned findings, our data shows that in addition to the prevalent *GJB2* p.V27I, p.E114G and p.I203T mutation sites in Taiwan. Further research is warranted to delve into the possibilities of divergent hearing loss severity associated with *GJB2* mutations.

## 3. Discussion

Most pathogenic mutations in the *GJB2* gene lead to extensive protein misfolding, which prevents connexin 26 (Cx26) from trafficking to the plasma membrane or assembling into functional gap junction plaques. No discrete mutational “hotspots” for single amino acid substitutions have been identified in Cx26 in association with non-syndromic sensorineural hearing loss (NSHL) [14]. Instead, single-point mutations are distributed relatively evenly across the protein. Among the four transmembrane domains, TM2 exhibits the highest mutation density (67%), whereas TM4 shows a lower density of approximately 40%, and the least affected domain demonstrates a density of 33%. These estimates remain approximate because the X-ray crystal structure lacks membrane representation, and the electron microscopy (EM) structure provides only limited resolution of the membrane bilayer [15,16,17].

The GJB2 p.I203T variant was first documented in Asia by Takayuki Kudo and colleagues in 2000, when it was identified during a genetic analysis of 39 children with congenital deafness. In the same year, an extended screening of 154 hearing-impaired children revealed a single individual harboring the p.I203T substitution, which was notably associated with profound hearing loss [18]. In 2007, Wu et al. in China reported *GJB2* p.I203T among newborns with hearing impairment [19], and Hwa et al. later documented the variant in Taiwanese patients [20]. Initially, many researchers regarded *GJB2* p.I203T as a benign polymorphism, and its clinical impact on hearing impairment received little further investigation. Reported allele frequencies vary regionally, from 0.3% in China to 5.4% in Japan, 1.5% in Korea, and up to 6.5% in Taiwan [21,22,23]. In 2013, 658 Chinese cases and found *GJB2* p.I203T in four patients—one with severe and three with profound hearing loss. In the present study, the allele frequency of *GJB2* p.I203T (6%) was consistent with prior Taiwanese data; however, its phenotypic expression varied: one carrier exhibited normal hearing and another profound loss, both aged > 18 years. These observations suggest that additional genetic or environmental modifiers may influence the severity of hearing impairment in *GJB2* p.I203T carriers, although such factors remain unidentified.

Our mutation frequency analysis also demonstrated high prevalence of *GJB2* p.V27I (31%) and *GJB2* p.E114G (22%) in Taiwan, compared with *GJB2* p.V37I (7.7%). Given the common occurrence of *GJB2* p.V27I and *GJB2* p.E114G, further research is warranted to determine their clinical relevance in hearing loss pathogenesis. Notably, the established pathogenic variant *GJB2* p.V37I is also relatively frequent in Taiwan, with an allele frequency of 6%. These findings raise important clinical considerations, as a substantial portion of the population could potentially harbor combinations of these variants, which may influence auditory outcomes.

Functional studies have begun to shed light on the biological effects of these variants. In 2016, Chen et al. generated a homozygous *GJB2* p.V37I knock-in (KI) mouse model using embryonic stem cell gene-targeting techniques [24]. While no gross morphological or developmental cochlear abnormalities were observed, confocal immunostaining and electron microscopy revealed mild outer hair cell loss. Auditory brainstem response (ABR) thresholds in homozygous KI mice were elevated relative to wild-type (WT) controls. Transcriptomic profiling identified altered expression of Fcer1g, Lars2, Nnmt, and Cuedc1 in KI mice [25]. Of these, Fcer1g encodes the γ chain of the high-affinity IgE receptor, integral to immune signaling [26]; Lars2 mutations are implicated in Perrault syndrome–associated hearing loss, likely through neural developmental deficits rather than direct cochlear pathology [27]; CUEDC1 suppresses epithelial–mesenchymal transition via the TβRI/Smad pathway and inhibits tumor progression in non-small cell lung cancer [28]. Nnmt, of particular interest to cochlear physiology, participates in homocysteine metabolism. Elevated Nnmt expression in *GJB2* p.V37I KI mice could influence S-adenosylmethionine (SAM) balance, impair homocysteine demethylation, and lead to increased homocysteine levels—an established risk factor for heightened reactive oxygen species (ROS) production [29].

Emerging evidence links hearing loss to an elevated risk of dementia. A recent longitudinal study demonstrated that even mild hearing loss doubles the likelihood of cognitive decline, while moderate and severe losses increase the risk approximately threefold and fivefold, respectively, over more than a decade [30].

Genotype analysis of 23 patients in our cohort showed that *GJB2* p.V37I and *GJB2* p.I203T occur on different chromosomes and do not co-segregate; no individual carried both mutations. Given the short physical distance between the loci (~500 bp within the intronless *GJB2* coding region), the co-occurrence of both mutations in cis would be rare, except in the event of crossover within this interval. Consequently, the probability of identifying hearing-impaired individuals in Taiwan harboring both mutations is expected to be extremely low.

We postulate that mutations in *GJB2* may contribute to age-dependent accumulation of reactive oxygen species (ROS), thereby potentiating the severity of hearing impairment. Nonetheless, such an effect was not discernible within the present cohort. Several limitations warrant consideration: (i) the mere number of GJB2 variants does not appear to correlate directly with the degree of hearing loss, and (ii) the relatively limited sample size constrains the statistical power of our analyses, highlighting the necessity for larger, well-characterized cohorts to more precisely delineate genotype–phenotype correlations and elucidate the underlying molecular mechanisms.

In summary, thanks to the development and cost reduction of today’s genetic sequencing technology, we recommended that clinical hearing loss gene screening should include whole-gene sequencing of *GJB2* and we propose that in addition to the well-established *GJB2* mutations as a cause of hearing loss, it is also important to consider whether interaction of *GJB2* with other genes, such as *ALDH2*, *NQO1*, *TXNRD1*, *PRDX4* and so on, may be involved in intracellular ROS metabolism [31], resulting in more severe hearing loss or compensatory effects. Cellular experiments and animal models targeting double mutations in, *GJB2* genes may lead to new discoveries in this field.

## 4. Materials and Methods

### 4.1. Patients and Methods

#### 4.1.1. Ethical Approvals

This study was conducted following the ethical principles of the declaration of Helsinki guiding medical research and accordance with the Human Body Research Act and the Ministry of Health and Welfare’s guidelines, as outlined in the document titled “Precautions for the Collection and Use of Human Body for Research” (Medical Journal No. 0950206912, dated 18 August 2006). This study received approval from the MacKay Memorial Hospital (IRB No. 23MMHIS325e). Prior to enrollment of the participants, the protocol was explained in the participants’ preferred language and sign language interpretation used for affected individuals for full comprehension, signed and verbal consent obtained before recruitment. Informed consent was obtained from all participants involved in all study protocols. The *GJB2* mutation spectrum was analyzed in 1517 Taiwanese patients using next-generation sequencing data sourced from the Taiwan Biobank (https://www.biobank.org.tw/, accessed on 21 August 2025).

#### 4.1.2. Patient Recruitment

We collected the data of 240 hearing loss patients and isolated genomic DNA and performed Sanger sequencing for 82 hearing loss patients at Mackay Memorial Hospital. Among them, we selected 23 cochlear implant (CI) cases to do whole-exome sequencing was performed (Table 1). After excluding individuals with syndromic hearing loss or additional cognitive or psychological impairments, genomic DNA was isolated and subjected to Sanger sequencing (Figure 4). Mucosal cells were collected using an oral scraper, and DNA purification was carried out using the Presto™ buccal swab DNA Extraction Kit (Geneaid, New Taipei City, Taiwan). Cells were lysed and proteins degraded using Proteinase K and chaotropic salts, facilitating DNA binding to the fiberglass matrix of the spin column. Carrier RNA was employed to enhance DNA binding efficiency to the spin column membrane. Contaminants were removed using W1 and wash buffers containing ethanol. The purified genomic DNA was eluted with water at 60 °C. Further purification and quantification of genomic DNA were achieved using ultraviolet spectrophotometry (A260/A280).

A total of 240 patients (126 males and 114 females) with sensorineural hearing loss were initially enrolled. Genomic DNA was extracted from all individuals. Among them, 82 patients underwent Sanger sequencing for preliminary genetic screening. In general, each sample was sequenced once; however, in cases where base calling was ambiguous or variants were difficult to interpret, the Sanger sequencing procedure was repeated one to two additional times to ensure accuracy. From this cohort, 23 patients (13 females and 10 males) who had undergone cochlear implantation (CI) were subsequently selected for whole-exome sequencing (WES). Individuals with syndromic hearing loss or additional cognitive or psychological disorders were excluded from further analysis prior to genetic testing.

#### 4.1.3. Audiological Assessments

Audiological assessment was conducted using the auditory brainstem response (ABR) test under sedation to determine hearing thresholds at 2 and 4 kHz, following the methodology described by [32]. The hearing level (HL) of the better ear was assessed at 2 and 4 kHz using BioLogic equipment (https://www.biologic.net/, accessed on 12 October 2021). In brief, electrodes were positioned on the patient’s head in preparation for the ABR test. These electrodes were attached to the patient’s skin and connected to a computer, which recorded brainwave activity in response to auditory stimuli delivered through headphones. Throughout the procedure, patients remained at rest or asleep and were not required to provide any responses or perform any actions. In our study, hearing loss was categorized using the American Speech-Language-Hearing Association (ASHA) classification for “type, degree, and configuration of hearing loss” The classification was as follows:Normal hearing: −10 to 15 dB HLSlight hearing loss: 16 to 25 dB HLMild hearing loss: 26 to 40 dB HLModerate hearing loss: 41 to 55 dB HLModerately severe hearing loss: 56 to 70 dB HLSevere hearing loss: 71 to 90 dB HLProfound hearing loss: >91 dB HL

#### 4.1.4. GJB2 Mutations, PCR, and Sanger Sequencing

The *GJB2* gene spans 5513 base pairs (bp) and consists of two exons, measuring 193 bp and 2141 bp, respectively, with an intervening intron of 3179 bp. The mRNA transcript of *GJB2* is 2334 bp in length (GenBank NM_004004.5). Within exon 2 of *GJB2*, there is a 678-bp open reading frame which encodes the Cx26 protein. An exploration for alternative transcription initiation sites across the genome revealed an additional 184-bp exon within the intron, as indicated by an expressed sequence tag (EST) (GenBank DA975033.1) derived from a cDNA library of synovial tissue from individuals with rheumatoid arthritis [33]. Further investigation is required to determine if this mRNA variant is expressed in the inner ear. The presence of this alternative additional exon suggests a potential new focus for genetic screening in non-syndromic hearing loss (NSHL) patients [34].

For the detection of *GJB2* c.109G>A (p.V37I) and c.235delC (p.L79CfsTer3) mutations, forward and reverse primers (*GJB2*-F: CCTCATCCCTCTCATGCTGT

*GJB2*-R: TGCTTGCTTACCCAGACTCA, product size: 837bp) were employed in PCR amplification using Taq DNA polymerase. The PCR products were evaluated via 2% agarose gel electrophoresis and subsequently sequenced using Sanger sequencing (Genomics Inc., New Taipei City, Taiwan). Sequence variations were identified by comparison with *GJB2* reference sequences from GenBank accession numbers M86849, U43932, or XM_007169. The adenine nucleotide of the translation initiation codon ATG in exon 2 was designated as +1. Pairwise alignment with reference sequences was conducted using SnapGene^®^ (version 8.1.1, running in Viewer mode) software.

#### 4.1.5. GJB2 Whole Exome Sequencing

Genomic DNA was isolated from frozen peripheral venous blood using a commercial DNA extraction kit (Invisorb Blood Universal Kit 1000, STRATEC Molecular, Berlin, Germany). In addition to the screening of *GJB2* variations by traditional Sanger sequencing, as previously described, all patients were selected for trio analysis by whole-exome sequencing (WES). Approximately 100 ng of PCR products were sheared to approximately 120 base pairs by using M220 Focused-ultrasonicator (Covaris, Woburn, MA, USA). The library was generated using the KAPA HyperPrep Kit (Kapa Biosystems, Wilmington, MA, USA; a Roche company, Cape Town, South Africa) according to the manufacturer’s instruction. The quality of library was checked with 4150 TapeStation System (Agilent Technologies, Santa Clara, CA, USA) and Qubit dsDNA HS Assay Kits (Thermo Fisher Scientific, Waltham, MA, USA). The library was adjusted to a concentration of 20 pM and sequenced on the MiSeq with the MiSeq Reagent Kit v2 for 300 cycles (Illumina, San Diego, CA, USA) according to the manufacturer’s instructions.

#### 4.1.6. Statistical Analysis

The data are presented as means ± standard deviations (SDs). Nonparametric statistical analyses, including the Kruskal-Wallis test and Mann-Whitney U test, were conducted using Prism 6 software. Statistical significance was defined as a *p*-value less than 0.05.

## 5. Conclusions

In this study, we comprehensively analyzed the mutation spectrum of the *GJB2* gene in the Taiwanese population, highlighting four prevalent missense variants: *GJB2* p.V27I, *GJB2* p.V37I, *GJB2* p.E114G, and *GJB2* p.I203T. Our mutation frequency analysis also revealed a high prevalence of the *GJB2* p.V27I (31%) and *GJB2* p.E114G (22%) variants compared to the *GJB2* p.V37I (7.7%) variant in the Taiwanese population. Further investigation into the clinical significance of these two prevalent *GJB2* p.V27I and *GJB2* p.E114G variants is necessary to better understand their roles in disease manifestation and progression.

The *GJB2* p.I203T variant, although previously considered a polymorphism, may exert pathogenic effects and be associated with more severe hearing impairment. Moreover, structural modeling and population data suggest a genotype-phenotype correlation, with *GJB2* p.I203T potentially disrupting gap junction formation. These findings underscore the clinical importance of including comprehensive *GJB2* screening, especially for *GJB2* p.I203T, in genetic counseling and early detection programs in Taiwan. Future studies integrating functional assays and larger cohorts are needed to further elucidate the role of *GJB2* p.I203T and potential interactions with redox-related genes such as *ALDH2*. Such insights may contribute to the development of precision medicine approaches for hereditary hearing loss.

## Figures and Tables

**Figure 1 ijms-26-08213-f001:**
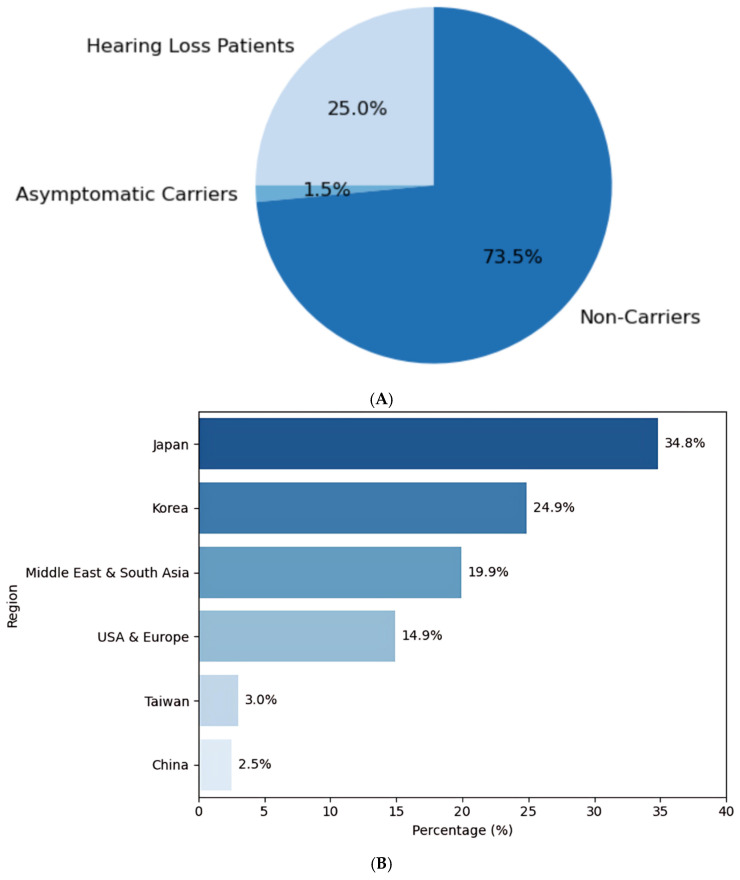
(**A**) Distribution of *GJB2* mutation carriers and hearing loss prevalence in the Taiwanese population. The pie chart illustrates the distribution of *GJB2* mutation status among a Taiwanese cohort. Approximately 25.0% of individuals with *GJB2* mutations present with hearing loss (Hearing Loss Patients), while 1.5% are asymptomatic carriers. The majority (73.5%) of the population are non-carriers. These data emphasize the clinical significance of *GJB2* mutations in contributing to non-syndromic hearing loss, as well as the importance of genetic screening in identifying carriers, even among asymptomatic individuals. (**B**) Comparative prevalence of *GJB2*-related hearing loss across global populations. This chart shows the relative prevalence of *GJB2*-related hearing loss among different geographic regions. Japan exhibits the highest reported prevalence (34.8%), followed by Korea (24.9%) and the Middle East & South Asia (19.9%). Taiwan accounts for 3.0% of reported cases, with lower frequencies observed in Mainland China (2.5%) and moderate representation from the USA & Europe (14.9%). These regional differences reflect population-specific genetic backgrounds and support the need for tailored screening strategies based on ethnic and geographic context.

**Figure 4 ijms-26-08213-f004:**
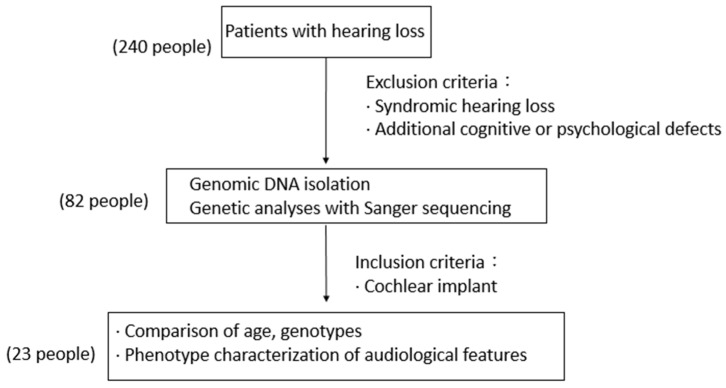
Summary of Patient Recruitment and Selection for Genetic Analysis.

**Table 1 ijms-26-08213-t001:** *GJB2* gene mutations in 23 cochlear implant patients total 13 females and 10 males.

Subject	Age	Gender	p.V27I	p.V37I	p.E114G	p.I203T	R’t PTA (dB HL)	L’t PTA (dB HL)	Other Mutation
1	10	Female	Het	WT	Het	Het	111.3	113.8	
2	38	Male	WT	Hom	WT	WT	90	106.3	
3	20	Female	Het	WT	WT	WT	115	90.3	CMV
4	17	Male	Hom	WT	Hom	WT	>100	>100	
5	18	Female	WT	WT	WT	WT	>100	>100	GJB2:c.301_303del (p.Glu101del) (+/−); GJB2:c.235del (p.Leu79fs) (+/−)
6	13	Male	WT	WT	WT	WT	102.5	101.3	suspect noonan syndrome with multiple letigenes or LEOPARD syndrome
7	24	Female	WT	WT	WT	Het	>100	>100	
8	44	Female	WT	WT	WT	WT	125	121.3	suspect MELAS syndrome X
9	58	Female	Het	WT	Het	WT	101.3	101.3	
10	12	Male	Het	Het	Het	WT	77.5	75	
11	8	Female	WT	Het	WT	WT	111.7	111.7	
12	7	Male	Het	WT	Het	WT	110	110	R’t Mondini dysplasia; L’t common cavity, cochlear nerve aplasia
13	53	Female	WT	WT	WT	WT	95	106.3	
14	14	Female	Het	WT	Het	WT	83.75	115	
15	23	Female	Het	Het	Het	WT	98.75	125	
16	11	Male	WT	WT	WT	WT	100	86.7	
17	2	Female	Het	Het	Het	WT	107.5	103.75	
18	11	Male	Het	WT	Het	WT	115	92.5	
19	16	Female	WT	WT	WT	WT	>100	>100	
20	5	Male	Het	Het	WT	WT	115	115	
21	11	Male	Het	WT	Het	WT	101.25	105	
22	10	Female	Het	WT	Het	WT	97.5	88.75	SLC26A4 (c.919-2A>G/WT) OTOF (c.5098G>C/WT)
23	18	Male	WT	WT	WT	Het	100	101.25	

## Data Availability

All data generated or analysed during this study are included in this article. Further enquiries can be directed to the corresponding author.

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
