# Peer review of "Mutation Spectrum of GJB2 in Taiwanese Patients with Sensorineural Hearing Loss: Prevalence, Pathogenicity, and Clinical Implications"

_ijms, 2025, doi:10.3390/ijms26178213_

Round 1
Reviewer 1 Report
Comments and Suggestions for Authors
Comments
The manuscript “Mutation Spectrum of GJB2 in Taiwanese Patients with Sensorineural Hearing Loss: Prevalence, Pathogenicity, and Clinical Implications”. The study aims to investigate the genotype-phenotype correlation of the four missense mutations with highest allele frequency of GJB2 gene in Taiwanese population. The genetic functional implications of hearing loss-related genes and clarify the pathogenesis of GJB2 related hearing loss and provide new insights and solutions for its prevention and protection. The paper is interesting. However, the points mentioned below should be improved.
Line 19-49: Correct it following journal format.
Line 40-44: Improve the sentence “ Although GJB2 mutations are the predominant genetic contributors to hereditary hearing loss, the mutant proteins display discernible aggregation, especially at cell–cell junctions, implying that they are at least partially transported to the plasma membrane. Genetic screening for these mutations, particularly GJB2 p.I203T (6%), p.V27I (31%), and p.E114G (22%), is crucial for managing NSHL in Taiwan.”
Line 114-137:How about the male/female or both samples? And the number of replicates for each sample? Explain clearly
Line 187: Results: It is better to explains in headings
Line 267: Figure 3, Currently known distribution of GJB2 gene mutations “here” add more details
Line 292-355: Please add clear pictures.
Line 420-424: Rephrase it “Our mutation frequency analysis also revealed a high prevalence of the GJB2 p.V27I (31%) and p.E114G (22%) variants compared to the GJB2 p.V37I (7.7%) variant in the Taiwanese population. Further investigation into the clinical significance of these two prevalent V27I and E114G variants is necessary to better understand their roles in disease manifestation and progression”.
Line 465-472:Rephrase it “ In summary, thanks to the development and cost reduction of today’s genetic sequencing technology, we recommended that clinical hearing loss gene screening should include whole-gene sequencing of GJB2 and we propose that in addition to the well-established GJB2 mutations as a cause of hearing loss, it is also important to consider whether interaction of GJB2 with other genes, such as ALDH2, NQO1, TXNRD1, PRDX4 and so on [30], may be involved in intracellular ROS metabolism, resulting in more severe hearing loss or compensatory effects. Cellular experiments and animal models targeting double mutations in, GJB2 genes may lead to new discoveries in this field”.
Comment: What is the novelty research in this study?
Line 485-644: Check the references following the journal format
Comment: Please improve the English, preferably with the help of a native speaker.
Comment: Revise the manuscript according to the journal's formatting guidelines.
Author Response
We sincerely appreciate your time and effort in carefully evaluating our manuscript and providing valuable comments and suggestions. These insights have greatly helped us improve the quality and clarity of the work. Please find our detailed responses below.
Comments 1: Line 19-49: Correct it following journal format.
Response 1: We sincerely appreciate your response. In accordance with the journal’s formatting guidelines, which specify the following structure:
- Introduction
- Results
- Discussion
- Materials and Methods
- Conclusions
we have revised the manuscript accordingly. The updated version is provided in the attached files.
Comments 2: Line 40-44: Improve the sentence “ Although GJB2 mutations are the predominant genetic contributors to hereditary hearing loss, the mutant proteins display discernible aggregation, especially at cell–cell junctions, implying that they are at least partially transported to the plasma membrane. Genetic screening for these mutations, particularly GJB2 p.I203T (6%), p.V27I (31%), and p.E114G (22%), is crucial for managing NSHL in Taiwan.”
Response 2: We sincerely appreciate your response. We have, accordingly, done as below :”Although GJB2 mutations represent the predominant genetic cause of hereditary hearing loss, the corresponding mutant proteins exhibit detectable aggregation, particularly at cell–cell junctions, suggesting at least partial trafficking to the plasma membrane. Genetic screening for these mutations—especially GJB2 p.I203T (6%), GJB2 p.V27I (31%), and GJB2 p.E114G (22%)—is essential for the effective diagnosis and management of non-syndromic hearing loss (NSHL) in Taiwan.”
Comments 3: Line 114-137:How about the male/female or both samples? And the number of replicates for each sample? Explain clearly
Response 3: We sincerely apologize for the confusion caused by the formatting issue, which made it unclear which part you were referring to in lines 114–137. To address your possible concerns regarding the questions, “How about the male/female or both samples? And the number of replicates for each sample? Explain clearly,” we have organized the relevant information as follows:
If referring to our recruited participants – the details are fully summarized in Figure 1. Summary of Patient Recruitment and Selection for Genetic Analysis and its corresponding figure legend.
If referring to the Taiwan Biobank dataset – we regret to inform you that, due to current access restrictions and the present status in Taiwan, the online database is temporarily unavailable for browsing or searching. Once the database becomes accessible again in the future, we will conduct the suggested search and provide you with the relevant information accordingly.
Regarding the genetic analysis of our recruited participants – in general, each sample was sequenced once; however, in cases where base calling was ambiguous or variants were difficult to interpret, the Sanger sequencing procedure was repeated one to two additional times to ensure accuracy.
All of the above information has also been incorporated into the revised manuscript, as provided in the attached files.
Comments 4: Line 187: Results: It is better to explains in headings
Response 4: We sincerely appreciate your response and we have modified it according to your suggestion.
Comments 5: Line 267: Figure 3, Currently known distribution of GJB2 gene mutations “here” add more details
Response 5: In response to your comment, we have added a schematic diagram of GJB2 mutations adapted from the referenced literature. This figure illustrates frameshift mutations, missense mutations, and nonsense mutations. Since the majority of variants identified in our cohort are missense mutations, the present manuscript places greater emphasis on discussing this mutation type. We sincerely appreciate your suggestion, which prompted us to include this figure to enhance the clarity and comprehensiveness of our work.
Comments 6: Line 292-355: Please add clear pictures.
Response 6: We have used professional software to improve the resolution.
Comments 7: Line 420-424: Rephrase it “Our mutation frequency analysis also revealed a high prevalence of the GJB2 p.V27I (31%) and p.E114G (22%) variants compared to the GJB2 p.V37I (7.7%) variant in the Taiwanese population. Further investigation into the clinical significance of these two prevalent V27I and E114G variants is necessary to better understand their roles in disease manifestation and progression”.
Response 7: We sincerely appreciate your response and we have modified it according to your suggestion.
Comments 8: Line 465-472:Rephrase it “ In summary, thanks to the development and cost reduction of today’s genetic sequencing technology, we recommended that clinical hearing loss gene screening should include whole-gene sequencing of GJB2 and we propose that in addition to the well-established GJB2 mutations as a cause of hearing loss, it is also important to consider whether interaction of GJB2 with other genes, such as ALDH2, NQO1, TXNRD1, PRDX4 and so on [30], may be involved in intracellular ROS metabolism, resulting in more severe hearing loss or compensatory effects. Cellular experiments and animal models targeting double mutations in, GJB2 genes may lead to new discoveries in this field”.
Response 8: We sincerely appreciate your response and we have modified it according to your suggestion.
Comments 9: Comment: What is the novelty research in this study?
Response 9: This study presents several novel contributions to the field of hereditary hearing loss genetics. Most notably, it provides the first comprehensive characterization of the GJB2 p.I203T variant in the Taiwanese population, challenging its previously assumed benign status. Through combined analysis of bioinformatic pathogenicity scores, allele frequencies, and patient clinical data, the study suggests that p.I203T may exert a pathogenic effect, potentially contributing to more severe auditory phenotypes.
In addition, the study leverages population-scale genomic data from the Taiwan Biobank and integrates it with targeted sequencing of hearing loss patients, allowing for a rare comparison between asymptomatic carriers and affected individuals within the same ethnic context. By identifying and comparing the prevalence of GJB2 variants—particularly p.V27I, p.V37I, p.E114G, and p.I203T—this research highlights the importance of ethnicity-specific screening approaches and demonstrates the clinical utility of full GJB2 sequencing in East Asian populations.
Finally, the investigation proposes a new hypothesis regarding the role of oxidative stress and gene–gene interactions (e.g., with ALDH2, NQO1) in modulating the penetrance and severity of GJB2-related hearing loss, thereby opening new directions for precision medicine research and potential therapeutic interventions.
If you think this paragraph should be added to the manuscript, please let me know.
Comments 10: Line 485-644: Check the references following the journal format
Response 10: We sincerely appreciate your response and we have modified it according to your suggestion.
Comments 11: Comment: Please improve the English, preferably with the help of a native speaker.
Response 11: We sincerely appreciate your response and we have modified it according to your suggestion.
Comments 12: Comment: Revise the manuscript according to the journal's formatting guidelines.
Response 12: We sincerely appreciate your response and we have modified it according to your suggestion.

Reviewer 2 Report
Comments and Suggestions for Authors
The manuscript focuses on the deafness-associated gene GJB2 mutation within the Taiwanese population. Using hearing loss patients and individuals from the Taiwan Biobank, this study investigates the prevalence and predicted pathogenicity of common GJB2 mutations. These findings may help with population-specific screening strategies.
Here, I have certain concerns:
- Line 219, I203T mutation tends to present with profound hearing loss. Where did this data come from? What statistical analysis was performed? What is the definition of hearing loss severity in this manuscript?
- The whole-exome sequencing (WES) was performed based on the method, but in line 117, the author stated whole-genome sequencing was performed. Please clarify.
- The authors may need to report the age and sex distribution for healthy individuals from the Biobank.
Author Response
We sincerely appreciate your time and effort in carefully evaluating our manuscript and providing valuable comments and suggestions. These insights have greatly helped us improve the quality and clarity of the work. Please find our detailed responses below.
Comments 1: Line 219, I203T mutation tends to present with profound hearing loss. Where did this data come from? What statistical analysis was performed? What is the definition of hearing loss severity in this manuscript?
Response 1: Thank you for pointing this out. We fully agree with your comment. We sincerely appreciate your valuable feedback, which has greatly helped us improve the quality and clarity of our work.
Between 2000 to 2023, we identified only seven published studies reporting on the GJB2 p.I203T variant:
GJB2 mutation spectrum in Inner Mongolia and its comparison with other Asian populations (Yuan et al.);
GJB2 mutation spectrum in 2063 Chinese patients with nonsyndromic hearing impairment (Dai et al.);
Analysis of trafficking, stability and function of human Connexin 26 gap junction channels with deafness-causing mutations in the fourth transmembrane helix (Ambrosi et al.);
Effectiveness of sequencing connexin 26 (GJB2) in cases of familial or sporadic childhood deafness referred for molecular diagnostic testing (Wu et al.);
Novel mutations in the connexin 26 gene (GJB2) responsible for childhood deafness in the Japanese population (Kudo et al.);
Molecular mechanisms and clinical phenotypes of GJB2 missense variants (Mao et al.); and
GJB2 mutation spectrum in Inner Mongolia and its comparison with other Asian populations (Yuan et al., different dataset).
Across these reports, clinical information on p.I203T is limited, and the variant has often been classified as a polymorphism. Nevertheless, in all cases where audiometric data were available, affected individuals carrying this variant presented with profound hearing loss. Our findings in the present cohort are consistent with these observations. Given that p.I203T ranks as the fourth most frequent GJB2 missense mutation in our Taiwanese dataset, we sought to further document its clinical correlation and gather additional evidence on its potential pathogenicity.
In our study, hearing loss severity was classified according to the 2023 American Speech-Language-Hearing Association (ASHA) guidelines for “Type, Degree, and Configuration of Hearing Loss”:
Normal: –10 to 15 dB HL
Slight: 16 to 25 dB HL
Mild: 26 to 40 dB HL
Moderate: 41 to 55 dB HL
Moderately severe: 56 to 70 dB HL
Severe: 71 to 90 dB HL
Profound: ≥91 dB HL
We did not perform a formal statistical association analysis between p.I203T and hearing loss severity due to the limited number of cases reported in the literature and in our cohort; instead, our conclusion is based on descriptive comparison with previously published data.
Comments 2: The whole-exome sequencing (WES) was performed based on the method, but in line 117, the author stated whole-genome sequencing was performed. Please clarify.
Response 2: We appreciate the reviewer’s careful observation. This was a typographical error in the manuscript. The correct method used was whole-exome sequencing (WES), and the text in line 117 has been revised accordingly.
Comments 3: The authors may need to report the age and sex distribution for healthy individuals from the Biobank.
Response 3: We regret to inform you that, due to current access restrictions and the present status in Taiwan, the online database is temporarily unavailable for browsing or searching. Once the database becomes accessible again in the future, we will conduct the suggested search and provide you with the relevant information accordingly.

Round 2
Reviewer 2 Report
Comments and Suggestions for Authors
Please put the dataset accessibility statement in the paper accordingly.
I don’t have any further comments.
Author Response
comment 1:Please put the dataset accessibility statement in the paper accordingly.
Response 1:We sincerely appreciate your response. The requested information has been included on page 37, lines 528–530 of the revised manuscript for your kind consideration.
comment 2:Figures and tables can be improved.
Response 2:We sincerely thank the reviewer for this insightful and constructive suggestion regarding the improvement of figures and tables.
In response, we have carefully standardized the formatting of both tables to ensure consistency across the manuscript. In addition, we have redesigned and unified Figure 1 to enhance its clarity and overall presentation. Furthermore, we have improved the resolution of all figures to the maximum extent possible within our technical capacity.
We are truly grateful for this valuable comment, which has allowed us to refine the quality, readability, and overall presentation of our manuscript.
